# Bioinspired crowding directs supramolecular polymerisation

Nils Bäumer[1], Eduardo Castellanos[2], Bartolome Soberats [2] &
Gustavo Fernández [1] ✉

Crowding effects are crucial to maintaining functionality in biological systems, but little is known about their role in analogous artificial counterparts. Within the growing field of supramolecular polymer science, crowding effects have hitherto remained underappreciated. Herein, we show that crowding effects exhibit strong and distinct control over the kinetics, accessible pathways and final outcomes of supramolecular polymerisation processes. In the presence of a pre-formed supramolecular polymer as crowding agent, a model supramolecular polymer dramatically changes its self-assembly behaviour and undergoes a morphological transformation from bundled fibres into flower-like hierarchical assemblies, despite no co-assembly taking place. Notably, this new pathway can only be accessed in crowded environments and when the crowding agent exhibits a one-dimensional morphology. These results allow accessing diverse morphologies and properties in supramolecular polymers and pave the way towards a better understanding of high-precision self-assembly in nature.

Biological entities, such as cells, require a wide variety of macro-molecules such as sugars and proteins to maintain their intricate functionality[1]. As these macromolecules are not present at high concentrations, biologists have termed these conditions as *crowded* rather than concentrated[2,3]. Despite that no specific interactions occur between most biomolecules within a cell, the shear presence of other (macro)molecular structures can have a strong non-linear impact on key processes, such as protein folding and enzyme-substrate association[2,4]. Accordingly, natural assemblies must efficiently maintain their intricate functionalities in crowded, multicomponent environments[2,5]. The chaperonin GroES-GroEL represents an archetypal example of how crowding effects govern physiological functionality (Fig. 1a)[6]. Under in vivo conditions, i.e. in crowded environments, it can efficiently promote folding processes to aid enzymes to reach their functional tertiary structure[7]. In contrast, under isolated conditions (in vitro), this functionality is inefficient, as the complex formation and substrate binding rely on the presence of crowding agents in solution[7]. Macromolecular crowding has also been linked to amyloidogenesis and protein fibrillation, which has tangible

implications for various diseases such as Alzheimer's or Parkinson's[8–10]. The fibrillation process[11,12] can be accelerated or decelerated by the presence of macromolecular crowders based on a counterplay between the influence of the excluded volume and the changes in viscosity[10,13]. Additionally, non-specific interactions between macromolecular crowders and proteins are also associated with reduced risks of dialysis-related amyloidosis in patients with high levels of serum albumin and the onset of Alzheimer's disease in vivo with reduced levels of serum albumin[14,15].

Supramolecular polymers can be considered as simplistic artificial model systems that replicate multiple key properties of biological assemblies, such as reversible binding, self-healing and diverse hierarchical organizations[16–18]. For these systems, recent focus has been placed on understanding supramolecular polymerisation processes in multicomponent mixtures[16,17,19,20]. These approaches have specifically aimed at controlling the morphology and monomer sequence in co-assembled structures, making use of social self-sorting[21]. However, although crowding effects have been examined in the context of crystallization processes and macromolecular systems, their role in

[1]Westfälische-Wilhelms Universität Münster, Organisch Chemisches Institut, Corrensstraße 36, 48149 Münster, Germany. [2]Department of Chemistry, University of the Balearic Islands, Cra. Valldemossa, Km. 7.5, Palma de Mallorca 07122, Spain. ✉e-mail: fernandg@uni-muenster.de

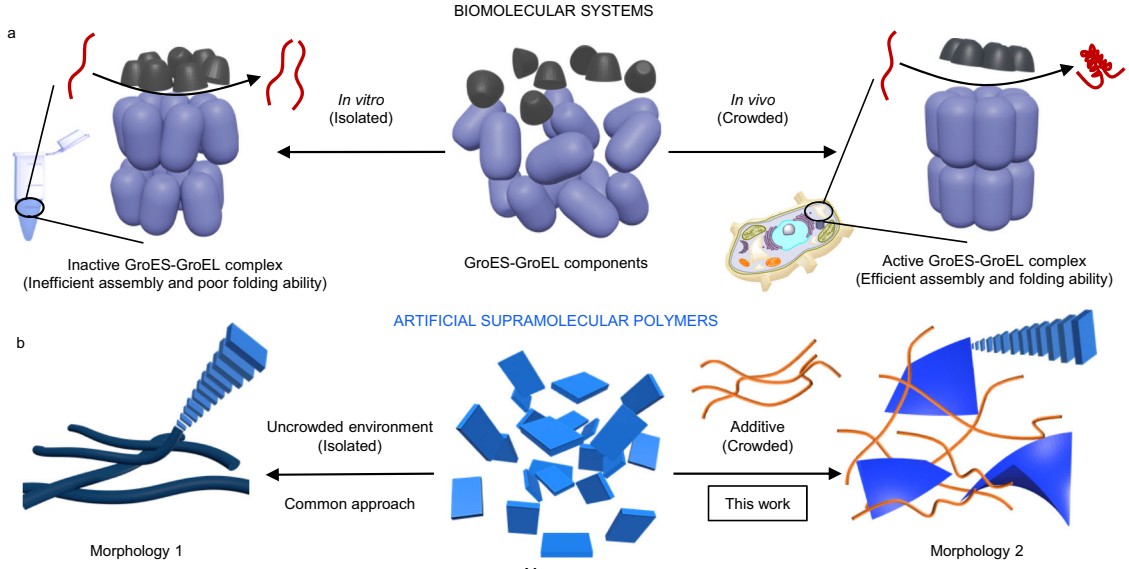

**Fig. 1 | Conceptual representation of crowding effects in biological and artificial systems. a** schematic representation of the GroES-GroEL complex assembled from 21 individual components (middle) in vitro (left) and in vivo (right), highlighting the influence of macromolecular crowding agents. **b** schematic representation of the distinct self-assembly behaviour of a model monomer (middle) in the absence (left) or presence (right) of a supramolecular polymer as crowding agent.

supramolecular polymers has thus far remained unexplored[3,22,23]. Here, we show that crowding effects control the kinetics, accessible pathways and final outcome of supramolecular polymerisation processes. To probe this concept, we treated a model supramolecular polymer that forms via fast kinetics with a second supramolecular polymer as crowding agent. We found that, although the crowding agent assembles independently, it simultaneously governs the supramolecular polymerisation process and opens up a new pathway into flower-like microstructures, which ultimately evolve into hierarchical nanoribbons. This new pathway can only be accessed in crowded environments and when the crowding agent exhibits a one-dimensional morphology, whereas the specific molecular structure of the crowding agent has no influence on the self-assembly outcome, enabling the use of diverse compounds as crowding agents.

Our results highlight that crowding effects direct the outcome of supramolecular self-assembly, enabling accessibility to more diverse morphologies without the need to change external parameters, such as solvent or temperature[24,25]. We anticipate that relevant functional properties gated behind specific (hierarchical) supramolecular morphologies will become accessible based on supramolecular crowding approaches[26–30], which should accelerate the development of smart materials based on functional supramolecular polymers[31–35].

## Results and discussion

### Preliminary considerations
In order to address crowding effects in self-sorted supramolecular systems, some key requirements need to be fulfilled. Firstly, two monomer units capable of forming supramolecular polymers (one acting as crowding agent and the second one as model supramolecular polymer, Fig. 1b) need to be selected. Secondly, both components must be able to self-assemble independent from each other, i.e. undergo narcissistic self-sorting, to avoid any potential influence of co-assembly processes. In addition, surface accelerated secondary nucleation processes must be avoided, for example using non-complementary morphological properties, i.e., stiff *vs.* flexible or globular *vs.* planar[36,37]. Finally, the crowding agent needs to self-assemble within a timeframe in which the model supramolecular polymer exists in a (kinetically trapped) monomeric state.

### Self-assembly in crowded environments
To probe this concept, we selected two model supramolecular synthons from our research group that satisfy the above-mentioned criteria: a linear Pt(II) complex (**1**) comprising two *trans*-arranged monodentate pyridine ligands and a V-shaped Pt(II) complex (**2**) derived from a bidentate bipyridine ligand (Fig. 2a)[38]. In brief, the linear complex **1** self-assembles in a cooperative manner into rigid bundles of fibres in low polarity solvents (Agg**1U**; Fig. 2b). In contrast, the higher preorganisation of **2** arising from the planar bipyridine ligand enhances the aggregation propensity, leading to thin and flexible one-dimensional supramolecular polymers even at lower concentrations (Agg**2**; Fig. 2b). The distinct timeframes at which **1** and **2** undergo self-assembly in the same solvent system and temperature allow us to use Agg**2** as a crowding agent, which may subsequently affect the hierarchical self-assembly of **1** (Fig. 2c). Initially, we confirmed the narcissistic self-sorting of **1** and **2** under thermodynamic conditions, i.e. slow cooling rates and mechanical agitation (Supplementary Fig 1–3, Supplementary Note 1). Following these preliminary studies, the self-assembly protocol was modified under kinetic regimes, in analogy to biological systems[2].

To examine the kinetic aspects of the self-assembly, solutions of **1** in MCH were heated to 363 K for no less than 45 minutes prior to being rapidly cooled down to room temperature. The resulting solutions were subsequently subjected to time-dependent UV/Vis measurements (see methods section for additional details). Under these conditions, an unfavourable nucleation followed by a rapid aggregation process leading to large self-assembled architectures with poor colloidal stability can be appreciated (Fig. 2d, Supplementary Fig. 4)[38,39]. On the other hand, the presence of a supramolecular crowding agent (Agg**2**) dramatically affects the self-assembly process of **1**. The lag time observed during UV/Vis measurements increased from merely 40 minutes in the case of pure **1** ($20 \times 10^{-6}$ M) to over 1000 minutes in the presence of **2** at a concentration of $2.5 \times 10^{-6}$ M (Fig. 2e, f). An even more pronounced time delay of nearly three days is observed when the amount of crowding agent is increased to $5 \times 10^{-6}$ M. Interestingly, the final equilibrium is also affected by this change in environment. The poor colloidal stability of the aggregates in uncrowded conditions (Agg**1U**) leads to a nearly complete precipitation of all compound in solution, as evidenced from the strong depletion of the absorbance after 4200 minutes (Fig. 2d). In stark contrast, the absorbance after

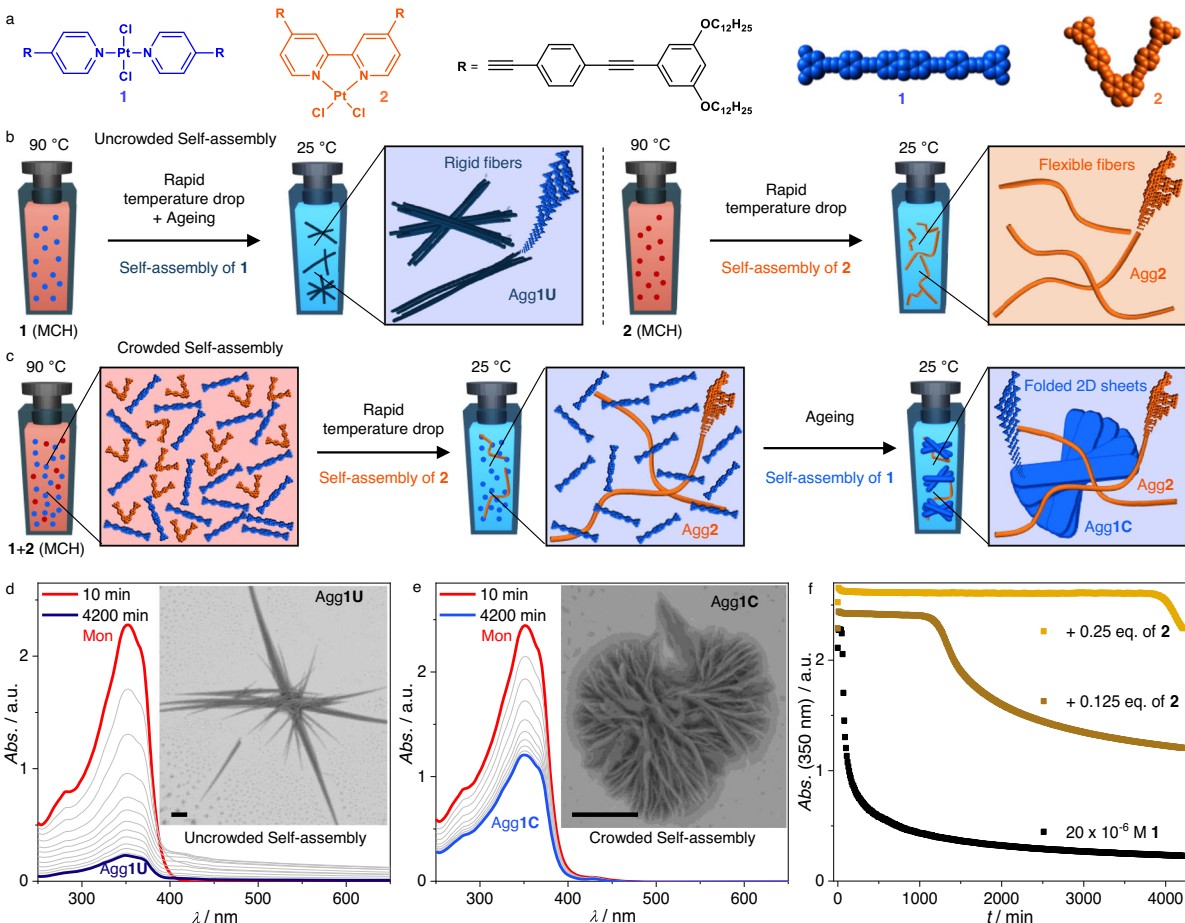

**Fig. 2 | Supramolecular polymerisation of 1 in uncrowded and crowded environments. a** chemical structures of **1** and **2**. **b** schematic representation of the self-assembly behaviour of **1** and **2** in isolation. **c** schematic representation of the self-assembly behaviour of **1** in a crowded environment. Time-dependent UV/Vis spectra of **1** (**d**) and **1** in the presence of **2** (**e**) with the corresponding supramolecular morphologies presented as insets (the scale bars correspond to 5 μm). **f**, time-dependent changes in the absorbance of mixtures of **1** ($c = 20 \times 10^{-6}$ M) and **2**, with increasing amounts of **2** ($c = 0$ (black), $2.5 \times 10^{-6}$ M (brown), and $5 \times 10^{-6}$ M (yellow)).

equilibration is only halved in a crowded scenario (Agg**1C**). The change in the final equilibrium was examined by microscopy techniques (insets Fig. 2d+e). The morphology of Agg**1C** observed by scanning electron microscopy (SEM) is best described as a complex hierarchical flower-like structure, which has been previously observed in the self-assembly of two-dimensional sheets[40–42]. This observation points to a new self-assembly pathway of **1** under crowded conditions, as the self-assembly in the absence of crowding agents solely leads to the formation of bundled one-dimensional fibres, irrespective of the concentration (see supporting information for details, Supplementary Fig 4-15 and Supplementary Note 2). Note that the flexible fibres of Agg**2** are not incorporated in the flower-like structures of Agg**1C**, but instead remain separately dispersed in solution, as demonstrated by atomic force microscopy (AFM), SEM and nuclear magnetic resonance (NMR) spectroscopy (Supplementary Fig 16-21 and Supplementary Note 3). Vigorous shaking of the assembled hierarchical flower-like structures enables partial disruption of the layered sheets and allows the visualization of individual sheets using AFM (Supplementary Fig 8). The individual height of these sheets ($h = 4.8$ nm) matches the molecular dimensions of **1** assuming a slightly tilted arrangement as a consequence of a small translational offset in the molecular stack.

To rationalize crowding effects in artificial systems compared to natural counterparts, some key differences must be taken into consideration. As a consequence of the large differences in total concentration of crowder (between 300 and 400 g/L in the case of *E. coli* for instance *vs.* below 1 g/L for the supramolecular polymers in this

study)[1,43], the influence of a supramolecular crowder cannot be solely rationalized by excluded volume effects. Further, the influence of viscosity should be evaluated on a case-by-case basis. In the present system, no changes in viscosity depending on the crowder concentration could be observed (Supplementary Table 1-4). Instead, the influence of an artificial supramolecular crowder should be traced to the potential to decrease the likelihood of the energetically unfavourable nucleation event, enhancing the lag phase of a kinetically controlled process by steric clashes[44–46]. In order to gain further insights into this potential mechanism of the crowding-controlled self-assembly, we have monitored the lag time of **1** by UV/Vis and photoluminescence studies in more detail revealing an unchanged spectroscopic behaviour in the presence and absence of Agg**2** (Supplementary Fig 22). In combination with the narcissistic self-sorting of both monomers and the absence of surface-catalysed secondary nucleation, we infer that the crowder exerts its influence based on repulsive interactions with **1**. Additionally, a potential deceleration of the growth process may lead to organizations of a higher hierarchical order, as has been observed for various supramolecular polymers and self-assembled architectures[47–49]. To further support this hypothesis, we conducted additional temperature-dependent experiments under kinetic conditions, in which the nucleation may be hindered, but an influence on the elongation can be neglected. To our satisfaction, an increase in the heating-cooling hysteresis could be observed in the presence of Agg**2**, while the final assemblies could clearly be characterized as Agg**1U** (Supplementary Fig 24 and Supplementary Note 4). Based on these results, we propose

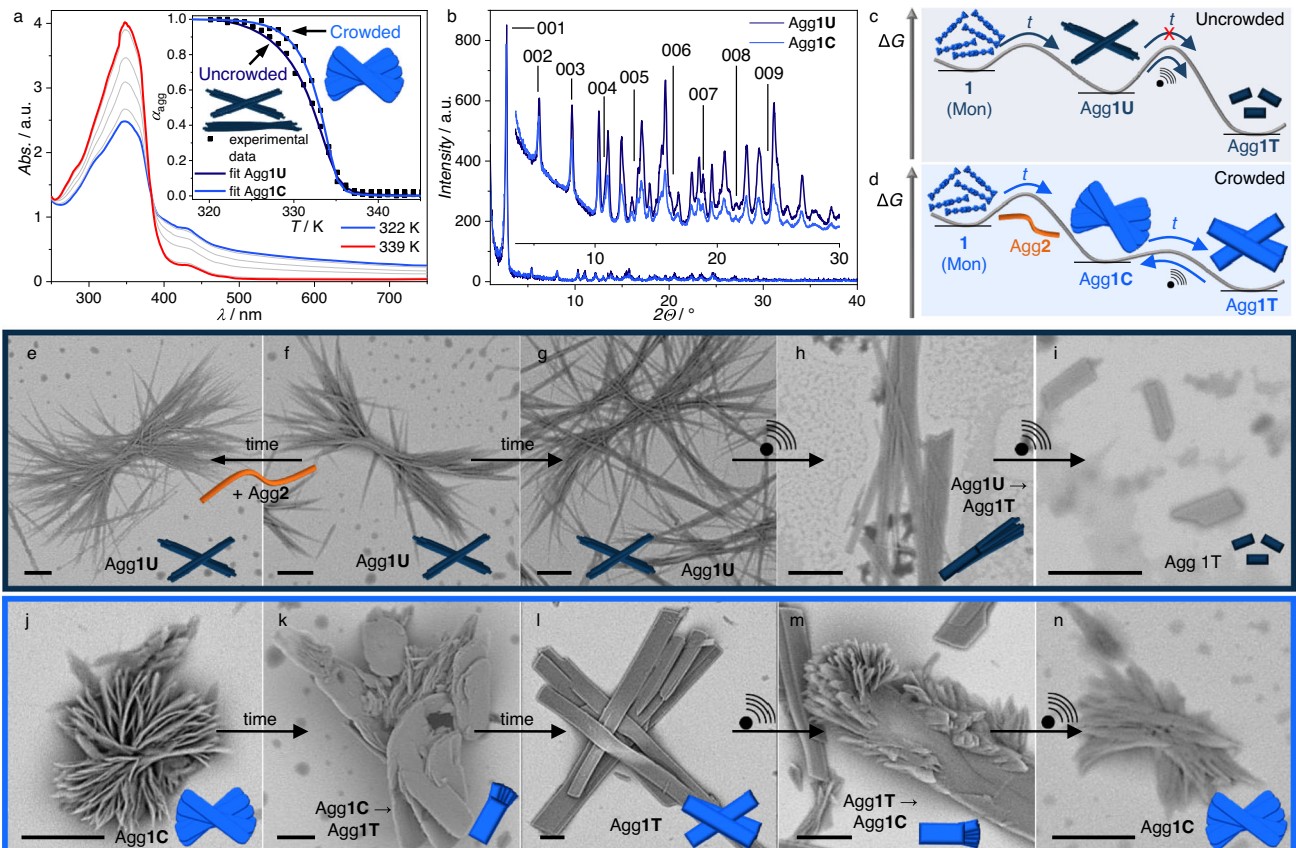

**Fig. 3 | Self-assembly behaviour of Agg1C. a** temperature-dependent UV/Vis spectra of **1** in crowded environments; inset: aggregation curves corresponding to the UV/Vis spectral changes in **a** with the corresponding aggregation curve in uncrowded environments for visual comparison. **b** XRD pattern of Agg**1C** and Agg**1U**. Qualitative energy landscape of the self-assembly of **1** in uncrowded (**c**) and crowded (**d**) environments. SEM micrographs of Agg**1U** showing the lack of structural evolution over time (**e**–**g**) and the structural evolution upon sonication (**h**, **i**). SEM micrographs of Agg**1C** showing the structural evolution over time (**j**–**l**) followed by prolonged sonication (**m**, **n**). The scale bars correspond to 5 μm. Changes in brightness and contrast have been equally applied to all SEM images for uniformity.

that crowding in our model supramolecular polymers is based on a dual mechanism, i.e. an increase in the lag time combined with a decrease in elongation rate, leading ultimately to the formation of assembled structures with a higher hierarchical order.

## Self-assembly pathways in crowded environments

In order to unravel the relative stability of Agg**1U** and Agg**1C**, we compared the thermodynamic parameters of both assemblies extracted from temperature- and solvent-dependent UV/Vis studies (for sample preparation and experimental details, see methods section). Temperature-dependent studies using controlled heating revealed a higher stability for Agg**1C** compared to Agg**1U** with regards to the degree of aggregation ($\alpha_{agg}$, Fig. 3a). This observation was consistent for various concentrations, while the elongation temperature ($T_e$) was identical for both aggregates for all investigated concentrations (Supplementary Fig 25–27)[50]. Based on the identical $T_e$ observed from the temperature-dependent studies, it can be inferred that the nucleation event is similar under crowded and uncrowded conditions. However, the change in hierarchical organisation of the individual stacks under crowded conditions leads to the formation of two-dimensional sheets with a higher hierarchical stability compared to the bundled fibres of Agg**1U**.

Additionally, the thermodynamic stabilities of Agg**1U** and Agg**1C** were investigated by denaturation experiments, using CHCl₃ as denaturing solvent[51,52]. These experiments disclosed nearly identical disassembly curves (Supplementary Figs. 28, 29), which underlines that the relatively higher hierarchical stability of Agg**1C** is not maintained during the addition of more polar chloroform. Interestingly, AFM analysis at intermediate solvent mixtures reveals a drastic change in morphology for Agg**1C**. At intermediate volume fractions of chloroform (16%) in MCH, a transformation of the two-dimensional sheets into lamellae of one-dimensional fibres is observed (Supplementary Fig. 31). This morphology bears resemblance to that observed for Agg**1U** under identical conditions, indicating a partial disruption of the hierarchical organisation of the morphologies (Supplementary Figs 30,31)[53]. This observation agrees with the hypothesis that both aggregates exhibit the same molecular arrangement wherein only the hierarchical organisation is affected. In order to further confirm this hypothesis, we performed X-ray diffraction (XRD) experiments of the two different assembled states of **1** (Fig. 3b). The XRD patterns obtained for Agg**1C** and Agg**1U** show identical lamellar packings with interlayer distances of ~33 Å. This distance matches well with the molecular length and the layering pattern observed in AFM experiments (Supplementary Fig 8). More importantly, this observation supports the conclusions drawn from temperature- and solvent-dependent studies indicating that both aggregates are built on the same molecular arrangement but differ in their hierarchical levels. Furthermore, during a standard stability assessment of both aggregates over time, a morphological evolution was observed only in crowded environments (Fig. 3e–g; Agg**1U** vs. Fig. 3j–l; Agg**1C**). A systematic variation of the ratio between **1** and the crowding agent, as well as their concentrations, revealed an intriguing time-resolved transformation process of Agg**1C**: at low concentrations ($c \leq 40 \times 10^{-6}$ M), the hierarchical flower-like structures undergo an apparent folding and rearrangement process into highly ordered

ribbons with sizes between 5 and 35 microns (Agg**1T**, Fig. 3j–l, Supplementary Fig 32–37, Supplementary Note 5). In stark contrast, no time-dependent transformation of Agg**1U** could be observed irrespective of the concentration, even after the addition of Agg**2** to preformed Agg**1U** (Fig. 3e–g). We also examined whether Agg**1U** undergoes a transformation into Agg**1C** or Agg**1T** using previously established mechanical agitation protocols (Fig. 3g–i)[54]. Notably, SEM analysis revealed an immediate conversion of Agg**1U** upon sonication into the nanoribbon structure observed after ageing solutions of Agg**1C**. This observation suggests that Agg**1U** represents a kinetically trapped topology in the energy landscape of **1**. Furthermore, this affirms that Agg**1T** represents the global energy minimum in the self-assembly of **1**, while both Agg**1C** and Agg**1U** are kinetic intermediates along hierarchical paths towards the same final structure (Fig. 3c, d)[55–57]. Considering the apparent similarity between Agg**1T** obtained after ageing Agg**1C** and the morphology observed after sonication of Agg**1U**, the stability of the aged product of Agg**1C** upon sonication was probed (Fig. 3l–n). Notably, short sonication periods did not lead to the fragmentation of the larger ribbons, but instead reversed the folding process to give the hierarchical flower-like structures observed immediately after self-assembly. We attribute this unexpected result to minor defects in the ideal ribbon structure, which, even if not observable by SEM, can lead to an efficient unfolding of the larger structures upon sonication.

The relatively high homogeneity of the nanomorphologies of Agg**1U** after sonication motivated us to investigate the seeded supramolecular polymerisation of **1** in uncrowded and crowded environments using the small ribbon structures as seeds (Fig. 4a, b; see methods section for a detailed description of the experimental protocol)[19,58,59]. These seeds maintain their structure over time and do not show any bias towards clustered agglomerates, which is a prerequisite for this approach (Supplementary Figs. 42)[60]. Addition of Agg**1T** seeds to aliquots of **1** in uncrowded conditions (Fig. 4a) leads to the formation of highly elongated morphologies resembling those of Agg**1U** (in the following termed Agg**1Us**), together with practically unchanged seeds (Fig. 4c and Supplementary Fig. 42). This observation suggests that the added seeds can serve as nucleation sites in uncrowded conditions, which then induces a highly efficient self-acceleration based on secondary nucleation effects[24,37,61,62]. Because of this inhomogeneous growth process, not all seeds act as nucleation sites due to the limited supply of **1**. Captivatingly, a drastically different seed-induced self-assembly behaviour is observed in crowded environments. After employing the seeded supramolecular polymerisation protocol, an efficient seed-induced polymerisation could be confirmed by SEM analysis, with the resulting microarchitectures resembling those of Agg**1T** (Agg**1Ts**). The sizes of these morphologies depend on the amount of added **1**, as expected (Fig. 4d and Supplementary Fig. 41). Thermodynamic analysis of the homogenous morphologies of Agg**1Ts** obtained in crowded environments reveals an intriguing size-dependent change in stability compared to Agg**1U** and Agg**1C**, reaffirming that the changes in stability of the different structures can be attributed to hierarchical effects (see Supplementary Note 6 and Supplementary Fig 38–46 for further details). These results highlight the importance of crowding effects in the production of structures with increased thermodynamic stability, both under kinetically controlled conditions and in the context of seeded supramolecular polymerisation.

## Scope and limitations
In order to probe the applicability of our method to other systems, we selected other available building blocks that fulfil the above-mentioned characteristics to act as supramolecular crowding agents, and subsequently conducted analogous studies under kinetic conditions (Fig. 5, Supplementary Fig 47–53)[63–65]. Initially, we focused on one-dimensional supramolecular polymers previously investigated in our group[63,64]. The results obtained using Agg**2** as

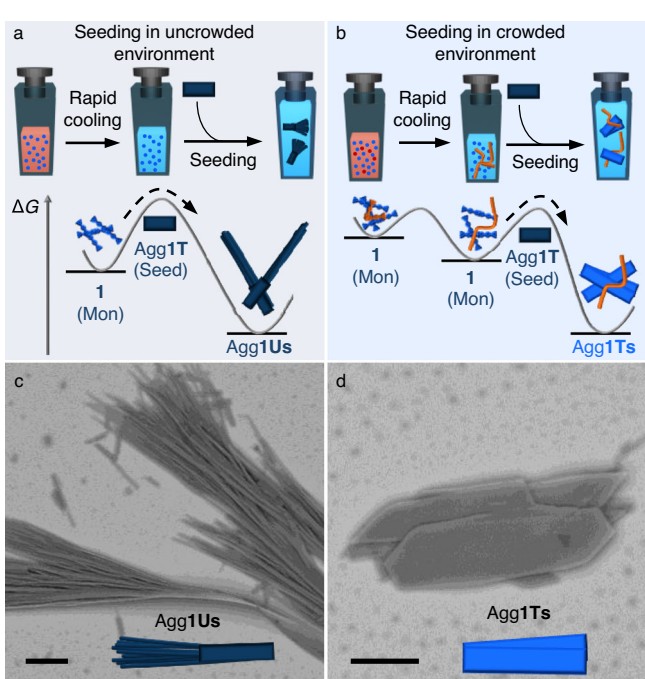

**Fig. 4 | Seeded supramolecular polymerisation in uncrowded and crowded environments. a, b** schematic depiction of the sample preparation protocol and energy landscape associated with seeded polymerisation in uncrowded (**a**) and crowded (**b**) environments. SEM micrographs of the supramolecular morphologies obtained from the experimental protocol illustrated in **a** (**c**) and **b** (**d**), respectively. The scale bars correspond to 5 μm.

crowding agent could be accurately reproduced using supramolecular synthons **3** and **4** (Fig. 5a, b; e, f). In both cases, an increase in the lag time prior to the self-assembly could be observed, ultimately producing the characteristic flower-like structures of Agg**1C**. Additionally, we used other aggregate morphologies, such as spherical nanostructures or two-dimensional nanosheets, as crowding agents[64,65]. These nanostructures do not affect the final aggregate morphology of **1** into fibre bundles (Agg**1U**), indicating that only elongated one-dimensional fibres are effective crowding agents for the present system (Fig. 5c, d, Fig. 5g, h). The same holds true for covalent polymers, which also failed to produce crowded environments, leading exclusively to the formation of Agg**1U**, even under highly elevated concentrations (see Supplementary Fig 47–53 and Supplementary Note 7 for details). On this basis, we conclude that the supramolecular morphology, rather than the specific molecular design, dictates the capabilities of a supramolecular polymer to act as a suitable crowding agent.

## Summary
In summary, we have demonstrated that binary systems undergoing narcissistic self-sorting can activate alternative self-assembly pathways induced by a crowded environment. This phenomenon bears close resemblance to examples of crowding phenomena in biological systems. Furthermore, we have shown that the potential of a supramolecular polymer to act as a crowding agent strictly depends on its morphology rather than its molecular design, which enables diverse supramolecular synthons to act as crowding agents. In addition to social self-sorting phenomena, our results disclose that crowding effects need to be considered in complex self-assembly environments. In the everlasting strife of supramolecular polymer researchers to rival the structural complexity of natural systems in terms of precision and functionality, these effects need to be acknowledged. We believe that our findings can serve as a foundation for future research and inspire diverse crowding-based

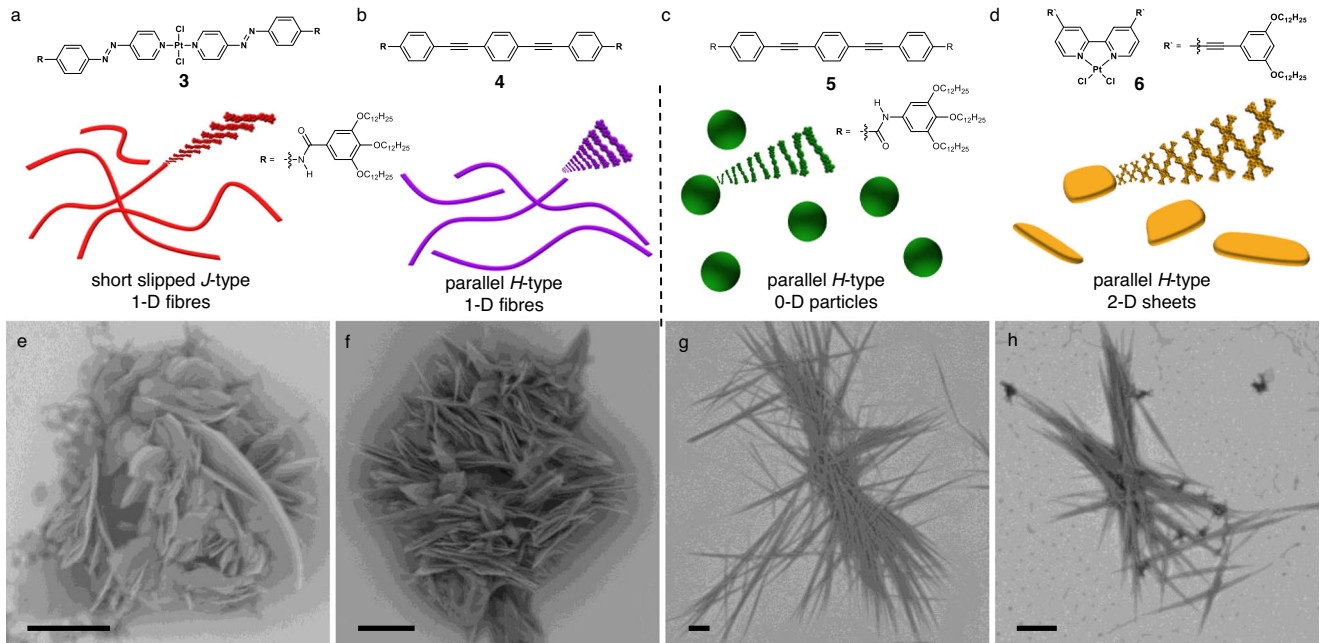

**Fig. 5 | Generalizability assessment of self-assembly induced by supramolecular crowding. a**–**d** chemical structures and schematic supramolecular self-assembly behaviour of crowding agents **3**, **4**, **5** and **6**. **e**–**h** SEM micrographs of Agg**1**C obtained in different crowding environments (see method section for experimental details); scale bars correspond to 5 μm.

approaches to broaden the scope of self-assembled topologies and resulting functional properties.

## Methods

### Synthesis
Compounds **1** and **2** have been synthesized using ligand replacement protocols from commercially obtained $PtCl_2(PhCN)_2$ in toluene. The respective ligands have been synthesized using a series of Sonogashira coupling reactions and deprotections using $K_2CO_3$ and tetra-n-butylammonium fluoride[38].

### UV/Vis spectroscopy
All UV/Vis spectra were recorded on a V-770, a V-750 and a V-730 spectrophotometer by the company JASCO with a spectral bandwidth of 1.0 nm and a scan rate of 400 nm min$^{-1}$. Glass cuvettes with an optical length of 1 cm were used for all measurements. All measurements have been conducted in solvents from commercial sources of spectroscopic grade.

### Atomic force microscopy
The AFM images have been recorded on a Multimode®8 SPM System manufactured by Bruker AXS. The used cantilevers were AC200TS by Oxford Instruments with an average spring constant of 9 N m$^{-1}$, an average frequency of 150 kHz, an average length of 200 μm, an average width of 40 μm and an average tip radius of 7 nm. All samples have been prepared using the protocol described below, unless specifically stated otherwise.

### Scanning electron microscopy
The SEM images were recorded on a Phenom Pharos Desktop SEM and on a Phenom ProX Desktop SEM manufactured by Thermo Fisher Scientific. The individual images have been recorded using a zoom between 22500× and 300× with either a BSD or SED detector and an acceleration voltage of either 5 or 10 kV (For individual images please see the corresponding figure caption). All samples have been prepared using the protocol described below, unless specifically stated otherwise.

### Dynamic and static light scattering
All DLS and SLS spectra have been recorded on a *CGS-3 Compact Goniometer System* manufactured by *ALV GmbH*, equipped with a HeNe Laser with a wavelength of 632.8 nm (22 mW) and an *ALV/LSE-5004 Digital Correlator* by *ALV GmbH*.

### X-ray diffraction experiments
X-ray diffraction (XRD) patterns were recorded on a BRUKER D8 Advance powder diffractometer (θ/θ geometry) with a nickel-filtered Cu-Kα radiation ($\lambda = 1.54$ Å).

### FT-IR spectroscopy
Thin film measurements were performed using a *JASCO-FT-IR-6800* with a CaF$_2$ window. Thin films were prepared by dropcasting 200 μL of the respective solutions in MCH followed by careful evaporation using an argon stream.

### Fluorescence spectroscopy
All fluorescence spectra were recorded on a *JASCO FP-8500* spectrofluorometer. Glass cuvettes with an optical length of 1 cm were used. All measurements were conducted in solvents from commercial sources of spectrophotometric grade.

### Rheological measurements
Rheological measurements were performed on an Anton Paar Modular Compact Rheometer MCR 102 (Anton Paar GmbH, Graz, Austria) with Anton Paar RhepCompass V1.20.40.496 (Anton Paar GmbH, Graz, Austria) analysis software. All measurements were conducted with a CP25-2 cone plate spindle (25 mm diameter) and a P PTD200 measuring cell. Measurements were performed with a constant shear rate of 50 /s at 21 °C.

### Kinetic self-assembly protocol
In order to investigate the kinetic self-assembly of **1** in crowded and uncrowded conditions, we freshly prepared solutions of the desired concentration by transferring the desired amount of compound from a stock solution in chloroform, followed by evaporation to dryness using

an argon stream. Subsequently, the necessary amount of MCH was added to achieve the desired concentration. For experiments in crowded environments, the required amount of crowding agent was added from a separate stock solution prior to solvent evaporation. As a standard protocol, the samples were heated to 368 K for no less than 45 min, but no more than 60 min to ensure a complete disassembly in the low polarity solvent and avoid any interference from potential seeds or small oligomers, which could perturb the experimental data. In order to induce the rapid temperature drop, the UV/Vis device was cooled to 298 K at the fastest cooling rate that the device is able to achieve by changing the setting to 298 K without intermediate steps (leading to an effective cooling rate of roughly 15 K/min). The time-dependent studies have been recorded over the course of 3 days using a measurement interval of 10 minutes.

### Temperature-dependent UV/Vis studies (heating)

In order to examine the thermodynamic stability of Agg**1C** and Agg**1U**, an initial stock solution of **1** at a concentration of $c = 40 \times 10^{-6}$ M was prepared in MCH following the established kinetic self-assembly protocol. For the preparation of Agg**1C**, 0.25 eq. of **2** was added prior to the kinetic self-assembly protocol. In order to investigate the disassembly process for multiple concentrations, the stock solutions were diluted using MCH and kept at 298 K for three days to ensure sufficient equilibration. All heating experiments have been recorded at a temperature range between 298 K and 368 K using a heating rate of 1 K/min and a measurement interval of 1 K. In order to minimize artifacts originating from the poor colloidal stability and high degree of clustering, the solutions were constantly stirred using a magnetic stirring bar at 1600 rpm. Thermodynamic analysis was performed using the nucleation-elongation model.

### Kinetic temperature-dependent UV/Vis studies (cooling)

In order to examine the effects of supramolecular crowding on kinetic cooling experiments, measurements were conducted after transferring aliquots of **1** or **2** from a highly concentrated stock solution ($c = 1 \times 10^{-3}$ M) to the measurement cuvette, followed by evaporation using an argon stream. After complete evaporation, the resulting solid was dissolved directly in MCH to give the final measurement solutions ($c_1 = 20 \times 10^{-6}$ M (uncrowded) and $c_1 = 20 \times 10^{-6}$ M; $c_2 = 5 \times 10^{-6}$ M (crowded)). Afterwards, the samples were heated to 343 K until both compounds were completely dissolved. The measurement was conducted with a cooling rate of 2 K/min without stirring and using a data interval of 2 K to minimize equilibration periods during the spectra acquisition.

### Solvent-dependent UV/Vis studies (denaturation)

In order to prepare the solutions of Agg**1C** and Agg**1U** for solvent-dependent studies, the sample preparation described in the previous section was used. Denaturation of the aggregates was achieved by the addition of aliquots of **1** dissolved in chloroform to gradually increase the solvent mixture polarity. The volume fraction of chloroform was increased in increments of 1% between each measurement. After addition of the required amount of chloroform, the cuvette was closed and gently inverted three times in order to minimize mechanical stress, while enabling sufficient solvent mixing. Following this procedure, one spectrum was recorded every 2 minutes, ensuring minimal contribution of solvent evaporation artifacts. For measurements of Agg**1C**, the chloroform stock solution contained the same equivalents of **2** as the aggregate solution to maintain the overall concentration of compound (**1** and **2**) present in solution. Thermodynamic analysis was performed using the denaturation model.

### Sample preparation for rheological measurements

Samples of Agg**2** in MCH were prepared by directly dissolving solid samples of **2** in MCH and heating to 363 K for 45 minutes for equilibration. After cooling to 298 K, the samples were kept at this temperature for 24 hours to ensure that the self-assembly of Agg**2** is under thermodynamic control. For each measurement, 300 μL of the respective sample were loaded onto the rheometer with a syringe and the plate was brought to a measuring distance of 0.106 mm. After trimming the sample with a wipe, an isolation hood was placed over the sample to maintain a constant temperature and avoid solvent evaporation.

### Sample preparation for microscopy

For all microscopy experiments, the same sample preparation protocol was employed in order to ensure adequate comparability between the different measurements, unless specifically stated otherwise. Before the solutions were coated onto the surfaces, the cuvette or vial containing the relevant sample was gently inverted 3 times to disperse the hierarchical flower-like structures in the solvent. Afterwards, 10 μL of the respective sample were dropcasted (HOPG for AFM, Si Wafer for SEM) and dried under ambient conditions. For the time-dependent studies on the morphological evolution, samples were prepared by following the time-dependent protocol described above and monitoring the progression of the self-assembly process by UV/Vis. Afterwards, the solutions were transferred into a small vial ($V = 2.0$ mL). The morphology was investigated before and after transferring the solutions to confirm that the morphology was unaffected. Afterwards, small aliquots ($V = 10$ μL) were used to conduct the SEM studies at different time intervals.

### Sample preparation for XRD experiments

The samples used for XRD analysis were obtained following the same experimental protocol described above for inducing kinetic self-assembly. In order to obtain enough sample, the supramolecular polymer was formed using 20 mg of **1** (plus 5 mg of **2** for crowded environments) dissolved in 100 mL MCH, which ensured a comparable concentration with previous UV/Vis experiments. Following the complete equilibration over a three-day period, the solvent was slowly removed using an Argon stream. Afterwards, the samples were dried *in vacuo* to remove residual solvent. The resulting powder samples were directly used for the XRD measurements, which were recorded under ambient conditions. The presence of the respective characteristic morphologies of Agg**1U** and Agg**1C** was confirmed by SEM experiments before and after the X-ray measurements (see Figure S31).

It is noteworthy that aggregate **2** is XRD silent, as previously reported[38]. In accordance with this, all the signals observed in the diffraction pattern of Agg**1C** correspond to **1**.

### Scope and limitation assessment

Different supramolecularly crowded environments were investigated by following the kinetic self-assembly protocol described above. For improved comparability, all measurements were performed using a $40 \times 10^{-6}$ M solution of **1**. At least two different concentrations were investigated for all crowding agents, according to the concentrations suitable for aggregation described in the respective original publications (for specific concentrations please refer to the individual figure captions in the additional data and supporting information). In order to avoid artifacts caused by small path lengths, a cuvette with a path length of 1 cm was used for all studies. In cases where the resulting samples exhibited absorbance levels beyond the range of the device, wavelengths other than the absorption maximum of **1** were used to evaluate the kinetic evolution.

### Seeded supramolecular polymerisation

The seeded supramolecular polymerisation studies were all executed following the same experimental protocol. Firstly, the specific aggregate used for seed formation was formed following the kinetic self-assembly protocol described above using a concentration of $40 \times 10^{-6}$

M. Secondly, the aggregates were subjected to sonication for a time frame of no less than one hour, with the progress of fragmentation being controlled at intermediate time frames by SEM. The aliquots of added **1** were freshly prepared for all studies using a heating period of no less than 45 min, followed by thermal quenching to 298 K. To initiate the seeded self-assembly, the seed solutions were mixed with the fresh aliquots of monomeric **1** in a 1:1 ratio. The corresponding time-dependent UV/Vis spectra were recorded immediately after seed addition. In order to prevent artifacts from spontaneous self-nucleation, the seeds were added without any lag time after the quenching protocol.

## Data availability

The data supporting the findings of this study are provided in the supporting information and are available from the corresponding author on request.

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

## Acknowledgements

N.B. and G.F. acknowledge the European Commission (ERC-StG-2016 SUPRACOP-715923) and the Deutsche Forschungsgemeinschaft (DFG, German Research Foundation)—GRK 2678—437785492 for funding. B.S. thanks the MCIU and AEI/FEDER of Spain for the projects EIN2020-112183 and PID2019-107779GA–I00, as well as for the "Ramón-y-Cajal" fellowship (RYC-2017-21789). We gratefully acknowledge Jonas Rickhoff (WWU Münster) for providing assistance with rheological measurements.

## Author contributions

N.B. and G.F. designed the project. N.B. performed the UV/Vis, FT-IR, photoluminescence, DLS, SLS, SEM and AFM studies. E.C. and B.S. performed and analysed the XRD experiments and provided the XRD section in the supporting information. N.B., B.S. and G.F. prepared the overall manuscript, including the figures. All authors contributed to the preparation of the manuscript by commenting and discussing the manuscript. The overall project was supervised by G.F.

## Funding

## Competing interests

The authors declare no competing interests.
