## [Peer Review File · Nature Communications]

Bioinspired Crowding directs Supramolecular PolymerisationREVIEWER COMMENTS

Reviewer #1 (Remarks to the Author):

Baumer et al. disclose the influence of crowding effects on supramolecular polymerization. The authors study how self-assembly of a linear Pt(II) complex is steered in the presence of other earlier formed Pt(II) supramolecular assemblies. They probe in detail the effect of a supramolecular polymer composed of a v-shaped Pt(II) complex on the self-assembly of the linear Pt(II) complex, revealing differences when performed under crowded (flower-like) vs. uncrowded (fiber) conditions. This concept is then further extended to other Pt(II) self-assemblies with varied morphologies, showing their influence on the self-assembly of the linear Pt(II) complex. Based on the enclosed findings that are of clear interest to the supramolecular polymer field, I recommend acceptance of the article in Nature Communications taking into consideration the following:

-The impact of macromolecular crowding on protein fibrillation processes such as amyloidogenesis has been well studied, but it is not considered in the text. Excluded volume effects and viscosity can influence amyloid fibril formation; the viscosity of the macromolecular crowder retards the aggregation process (e.g. J. Mol. Recognit. 2004, 17, 456; JBC 2012, 287, 38006). The authors see that addition of the crowder, and further increasing its concentration slows the aggregation of compound 1. Approximately doubling the concentration of 2 (2.5×10^{-6} to 5×10^{-6} μM) results in a lag time from 1000 min to 3 days. They point out that fibrillar crowdors 3, 4 and 7 are effective in altering the self-assembly process of 2 in a similar manner at different concentrations of crowder. Crowder 7 is a gelator at elevated concentrations. However, if crowdors 5 or 6 are used that are non-fibrillar there is no effect on the supramolecular polymerization of 2. Because of the results enclosed within this study and earlier reports in the field of protein fibrillation, it is critical for the authors to consider excluded volume effects and viscosity of the supramolecular crowdors in their analysis and discussion.

-In the introduction the description on macromolecular crowding in biology difficult to follow. Excluded volume effects and viscosity are not discussed that are involved in various intracellular processes involving proteins. Also, GroES-GroEL is highlighted here as an example and in the text a reference is made to protein folding and enzyme substrate association, but a stronger link to amyloidogenesis/protein fibrillation can be made based on the studies performed. The authors are recommended to revise this section as it is key to grasp the analogies that are being made for the fully artificial system. They may also want to highlight earlier on in the introduction of the manuscript the various Pt(II) supramolecular crowdors examined in the study, as the examination of their influence on the supramolecular polymerization of 1 is an interesting part of the work.

Reviewer #2 (Remarks to the Author):

I have mixed feelings about this paper. At one level, I like the fact that they are trying to propose a new concept in supramolecular chemistry: i.e., crowding effect. However, the goal is grand and the mechanism has not been convincingly supported. How does the crowding medium influence the supramolecular polymerization of 1? In biological systems, crowding effect is explained based on the excluded volume. Thus, the presence of background macromolecules (in the present case, 1D supramolecular polymer of 2) changes the accessible space for coexisting molecules (in the present case, compound 1) in the system. Do the authors expect something similar in the present system? Since the concentration of the crowding agents is low in comparison to those of macromolecules in biological systems, I assume other mechanism is operating here.

The authors repeatedly claim that crowding agents assembles independently and co-assembly does not take place. However, interaction between 1 and 2 is not completely ruled out under the kinetic condition. For example, 1 and 2 may interact within the lag time, which could also extend the lag time of 1 longer and eventually influence the morphology of the final outcome. Characterization of species formed in the lag time is missing.

As this study would open a new door in supramolecular polymer chemistry, I recommend this manuscript for publication in Nature Communications. However, the authors should discuss the mechanism at the molecular level with additional experimental evidences, particularly for the kinetic regime. In addition, as illustrated by GroES-GroEL complex shown in Figure 1a, it would be nice if the authors demonstrate that morphology 1 and morphology 2 of compound 1 are distinct in terms of their functions and properties, not only in terms of their morphology.

Reviewer #3 (Remarks to the Author):

The paper titled "Biomimetic crowding directs supramolecular polymerizations" has claimed that the one-dimensional supramolecular polymers could act as an effective crowder and influence the kinetics, thermodynamics and morphology of the supramolecular polymerization processes of the coexisted monomers. In particular, the new morphology or the flower like assemblies can only be seen with one-dimensional supramolecular polymers as crowdors, not with spherical or sheet like supramolecular polymers or covalent polymers. It is an interesting manuscript that can be recommended for publication, provided some changes are made. Based on this, there are a few questions below.

1. As the authors have compared the crowding effect in the supramolecular system with that in the biological system, could the authors relate the morphology of the crowdors to the excluded volume as what has been discussed in the biological system?

For structures with smaller excluded volume, could higher concentrations promote the formation of the flower like assemblies?

What's the molecular weight of the polystyrene tested in the paper? Could the polystyrene with a higher molecular weight promote the formation of the flower like assemblies?

2. Both the AFM and SEM are based on dried samples. How would the drying process influence the morphology? In the recently published paper "Hierarchically self-assembled homochiral helical microtoroids", micrometer scale hierarchical structures are formed by drop casting the pre-assembled intermediate colloids in solution. Could the formation of the flower like structures in this paper have been caused by the slow evaporation process? Could the samples be spin coated and be compared with the drop casted samples?

3. In the paper by Zimmerman and Minton (Annu. Rev. Biophys. Biomol. Struct, 1993), both of whom were cited by the authors, it was shown the reaction rate could first increase and then decrease with increasing amount of crowding reagent. If the concentration of the crowding reagent is reduced, could the acceleration of the supramolecular polymerization also be observed in addition to the delayed supramolecular polymerization observed in the paper?

4. In the abstract, the authors mentioned that "Within the growing field of supramolecular polymer science, crowding effects have hitherto not been considered". The crowding effect is rarely explored in the supramolecular polymer science, but there are still papers studying the crowding effect of covalent polymers on the self-assembly behavior of PIC (ref 3).

5. In Fig. 3a, the difference between Agg 1U and Agg 1C is small in the inset. Could an error bar be provided for both curves?

6. Fig. 5 what are the functional groups (R) in a and b?

Reviewer 1: Baumer et al. disclose the influence of crowding effects on supramolecular polymerization. The authors study how self-assembly of a linear Pt(II) complex is steered in the presence of other earlier formed Pt(II) supramolecular assemblies. They probe in detail the effect of a supramolecular polymer composed of a v-shaped Pt(II) complex on the self-assembly of the linear Pt(II) complex, revealing differences when performed under crowded (flower-like) vs. uncrowded (fiber) conditions. This concept is then further extended to other Pt(II) self-assemblies with varied morphologies, showing their influence on the self-assembly of the linear Pt(II) complex. Based on the enclosed findings that are of clear interest to the supramolecular polymer field, I recommend acceptance of the article in Nature Communications taking into consideration the following:

We sincerely thank the referee for supporting publication of our work and have addressed their concerns in the following manner.

1) The impact of macromolecular crowding on protein fibrillation processes such as amyloidogenesis has been well studied, but it is not considered in the text. Excluded volume effects and viscosity can influence amyloid fibril formation; the viscosity of the macromolecular crowder retards the aggregation process (e.g. J. Mol. Recognit. 2004, 17, 456; JBC 2012, 287, 38006). The authors see that addition of the crowder, and further increasing its concentration slows the aggregation of compound 1. Approximately doubling the concentration of 2 (2.5×10^{-6} to 5×10^{-6} M) results in a lag time from 1000 min to 3 days. They point out that fibrillar crowders 3, 4 and 7 are effective in altering the self-assembly process of 2 in a similar manner at different concentrations of crowder. Crowder 7 is a gelator at elevated concentrations. However, if crowders 5 or 6 are used that are non-fibrillar there is no effect on the supramolecular polymerization of 2. Because of the results enclosed within this study and earlier reports in the field of protein fibrillation, it is critical for the authors to consider excluded volume effects and viscosity of the supramolecular crowders in their analysis and discussion.

We appreciate the detailed comment and the very helpful link to other related literature on natural assemblies. Indeed, based on our experimental results, we infer that the molecular mechanism of crowding in our artificial system is not the same as in natural systems, as also suggested by referee 2. As we work under extremely dilute conditions, the influence of the excluded volume can be expected to be negligible (concentration of macromolecules in cells = 300 – 400 g/L; compared to the concentration of our supramolecular crowder < 1 g/L). As suggested by the referee, in order to investigate a potential influence of viscosity, we measured the viscosity of MCH depending on the amount of our supramolecular crowder 2, revealing an unchanged dynamic viscosity under conditions in which crowding effects of Agg2 were observed (see below and revised method section for experimental details). This data has also been added to the revised supporting information to support our proposed molecular mechanism.

Table S1: Summary of the rheology data obtained for a sample of pure MCH as reference (shear stress (σ), viscosity (η) and torque (τ)).

Run	T / K	σ / Pa	$\eta / mPa \cdot s$	$\tau / mN \cdot m$
1	294.06	0.03300	0.65994	1.3503E-4
2	294.13	0.03146	0.62915	1.2873E-4
3	294.15	0.03280	0.65607	1.3424E-4

Table S2: Summary of the rheology data obtained for a sample of Agg2 in MCH using a concentration of 10×10^{-6} M (shear stress (σ), viscosity (η) and torque (τ)).

Run	T / K	σ / Pa	$\eta / mPa \cdot s$	$\tau / mN \cdot m$
1	294.13	0.03633	0.68305	1.3976E-4
2	294.15	0.03415	0.69444	1.4209E-4
3	294.15	0.03472	0.65143	1.3329E-4

Table S3: Summary of the rheology data obtained for a sample of Agg2 in MCH using a concentration of 20×10^{-6} M (shear stress (σ), viscosity (η) and torque (τ)).

Run	T / K	σ / Pa	$\eta / mPa \cdot s$	$\tau / mN \cdot m$
1	294.15	0.03269	0.65372	1.3376E-4
2	294.15	0.03268	0.65350	1.3371E-4
3	294.15	0.03277	0.65539	1.3410E-4

Based on this comment as well as on the second comment by referee 1 and the first comment by referee 2, we have modified the introduction to make the analogies as well as the differences between artificial and natural systems clearer to potential readers. In addition, we have slightly adapted the title of our manuscript changing the word biomimetic to bioinspired, since although inspired by natural systems, there are some fundamental differences between crowding effects in biology and in our present system. As a result, we believe that the use of bioinspired is more appropriate than biomimetic for the current study.

2. In the introduction the description on macromolecular crowding in biology difficult to follow. Excluded volume effects and viscosity are not discussed that are involved in various intracellular processes involving proteins. Also, GroES-GroEL is highlighted here as an example and in the text a reference is made to protein folding and enzyme substrate association, but a stronger link to amyloidogenesis/protein fibrillation can be made based on the studies performed. The authors are recommended to revise this section as it is key to grasp the analogies that are being made for the fully artificial system. They may also want to highlight earlier on in the introduction of the manuscript the various Pt(II) supramolecular crowders examined in the study, as the examination of their influence on the supramolecular polymerization of 1 is an interesting part of the work.

We are thankful for the very helpful and constructive feedback from the referee. As already mentioned in response to the first comment, we have modified the introduction based on the reviewer's feedback. To this end, we first discuss the phenomenon of macromolecular crowding in natural systems in more detail and follow that with a transfer of the concept to artificial systems in general and our investigated system in particular. Additionally, we have also briefly mentioned the broad study of different crowding agents, as suggested by the referee.

Reviewer 2: I have mixed feelings about this paper. At one level, I like the fact that they are trying to propose a new concept in supramolecular chemistry: i.e., crowding effect. However, the goal is grand and the mechanism has not been convincingly supported. How does the crowding medium influence the supramolecular polymerization of 1? In biological systems, crowding effect is explained based on the excluded volume. Thus, the presence of background

macromolecules (in the present case, 1D supramolecular polymer of **2**) changes the accessible space for coexisting molecules (in the present case, compound **1**) in the system. Do the authors expect something similar in the present system? Since the concentration of the crowding agents is low in comparison to those of macromolecules in biological systems, I assume other mechanism is operating here.

We appreciate the concerns of the referee, which are largely in line with the comments made by referee 1. In response to these comments, we have expanded the introductory paragraphs and aimed to explain the molecular mechanism based on the experimental data at hand (please see also response to referee 1).

1) The authors repeatedly claim that crowding agents assemble independently and co-assembly does not take place. However, interaction between **1** and **2** is not completely ruled out under the kinetic condition. For example, **1** and **2** may interact within the lag time, which could also extend the lag time of **1** longer and eventually influence the morphology of the final outcome. Characterization of species formed in the lag time is missing.

We appreciate the very helpful comment from the referee. In order to address the species formed within the lag time of our experiments, we have performed additional (VT) UV/Vis analysis as well as additional fluorescence and light scattering studies to probe the nature of the (supra)molecular entities present within the lag time in more detail. Additionally, we have combined the insights from different AFM experiments in a more clear-cut manner to characterize the lag time regime. Due to the limited operational window of concentration in which we can observe the lag time ($c_1 < 40 \times 10^{-6} \text{ M}$), other approaches such as NMR are unfortunately not feasible.

Initially, we aimed to investigate the absorption properties of **1** within the lag time to gain more insights into the potential interactions with Agg**2**. To this end, we prepared dilute samples of individual **1** and **2** as well as their mixture ($c_1 = 20 \times 10^{-6} \text{ M}$; $c_2 = 5 \times 10^{-6} \text{ M}$) and confirmed the trapping of **1** in the molecularly dissolved state after conducting the kinetic protocol described in the experimental details. The UV/Vis spectra clearly indicate the absence of large assemblies, as can be inferred from the missing optical density in the low energy regime of the spectrum (please see spectra below). When comparing the absorption properties of **1** in uncrowded and crowded conditions, by subtracting the absorption of Agg**2**, it is clear that the only deviations in the spectra are present in areas where Agg**2** shows a relatively strong absorption (around 370 nm and above 400 nm), possibly owing to small errors in the sample preparation. In contrast, the position of the absorption maximum of **1** remains completely unchanged, indicating no strong chromophore-chromophore interactions in the ground state. As the excited state is generally more sensitive to changes in the chromophore environment, we subjected the same solutions to photoluminescence analysis. On the basis of these studies, it is apparent that the emission of **1** in crowded and uncrowded media remains unchanged, which further supports our assignment of the regime to the molecularly dissolved state of **1** with an energetically high nucleation barrier.

Figure 1: Top: UV/Vis spectra of **1** within the lag time in the absence and presence of Agg2, with the UV/Vis spectrum of Agg2 shown for comparison (left) and the normalized UV/Vis spectra of **1** in the absence and presence of Agg2. Bottom: Photoluminescence spectra of **1** in the absence and presence of Agg2 with the spectrum of Agg2 shown for comparison. ($c_1 = 20 \times 10^{-6} \text{ M}$; $c_2 = 5 \times 10^{-6} \text{ M}$; $\lambda_{\text{Ex.}} = 340 \text{ nm}$).

It is known in the literature (*Angew. Chem. Int. Ed.* **2020**, *59*, 19841; *J. Am. Chem. Soc.* **2021**, *143*, 11777) that interactions between molecularly dissolved species and the surface of a supramolecular polymer can occur, even if the thermodynamically most stable state is a narcissistically self-sorted system. These interactions have been linked to secondary nucleation processes, which accelerate an aggregate formation in the presence of other polymers. Although our observations are in direct contradiction with this behavior, we nevertheless sought to investigate whether **1** could nucleate on the surface of Agg2. To this end, we employed DLS studies. Unfortunately, the experimental results were inconclusive owing to the thin morphology and insufficient concentration, which prevents the observation of a clear autocorrelation function (please see below).

Figure 2: DLS correlation function of Agg2 using a concentration of $5 \times 10^{-6} \text{ M}$ in isolation and in the presence of **1** ($c_1 = 20 \times 10^{-6} \text{ M}$).

We also wish to reemphasize the time-dependent AFM studies represented in Fig. S18 of our initial submission (please see below). It can be clearly observed that during the lag time (a) only Agg2 can be visualized on the HOPG surface. Afterwards (within the nucleation regime), the formation of two-dimensional plates can be observed, which exhibit significant clustering upon elongation (b,c) suggesting the narcissistic self-assembly of **1**. Following the precipitation of the flower-like assemblies, the isolated Agg2 can be observed again. Crucially, these structures observed after precipitation of Agg1C are morphologically identical to those observed within the lag time of **1**, highlighting that no accumulation of **1** on the aggregate surface occurs. The same holds true for the fibers of Agg2 observed within the lag time of **1** under various conditions (see for example Fig. S5e, S6f).

Figure 3: AFM images revealing the narcissistic self-sorting of **1** and **2** over time, following the kinetic self-assembly protocol described in the methods section. a, AFM height image of Agg2 obtained shortly after the thermal quenching. b, AFM height image revealing small plates of Agg1C together with Agg2 obtained shortly after the self-assembly of **1** started. c, representative AFM phase image of Agg1C and Agg2 obtained after a short equilibration period (prior to precipitation of large flower-like structures of Agg1C). d, AFM height image of Agg2 obtained after full equilibration (precipitation of the flower-like structures of Agg1C). $c_1 = 40 \times 10^{-6} \text{ M}$; $c_2 = 5 \times 10^{-6} \text{ M}$.

We further investigated if an effect of the supramolecular crowder on the assembly of **1** could be achieved under fast cooling conditions ($\geq 2 \text{ K/min}$ cooling rate, 2 K data interval), i.e. intermediate experimental conditions between the kinetic self-assembly protocol described in the methods section and the cooling under thermodynamic equilibrium (Fig. S1). To our satisfaction, a minor increase in the heating/cooling hysteresis is observed in crowded media, although ultimately

the influence of the crowder is overshadowed by the continuous cooling enforcing a fast elongation of Agg1U (please see below).

Figure 4: VT UV/Vis spectra of **1** in the absence (top left, $c_1 = 20 \times 10^{-6}$ M) and in the presence of Agg2 (top right, $c_1 = 20 \times 10^{-6}$ M; $c_2 = 5 \times 10^{-6}$ M) with the changes in α_{agg} plotted against the temperature (bottom left) using fast cooling rates (2 K/min and 2 K data interval) for multiple repeated measurements. Bottom right: SEM micrograph of Agg1U formed in the presence of Agg2 under fast cooling conditions.

These results further support our original assumption that **1** exists as a molecularly dissolved species within the lag time, irrespective of whether this condition is achieved by dilution (uncrowded) or by the addition of a supramolecular crowder. This suggests that the supramolecular crowder can disfavor the formation of the nucleus, in a similar way as a simple dilution. In contrast to the diluted state, the crowded state can also suppress a one-dimensional growth, which we ascribe to repulsive interactions as a consequence of motion within the solvent. As a result, a two-dimensional growth behavior is facilitated, which ultimately leads to the formation of the flower-like particles. Based on our newly gathered experimental evidence as well as the results disclosed in our earlier analysis of the self-assembly of **1** (ref 38), we attribute the lag time and the hindered nucleation of **1** to the high flexibility of the solubilizing chains in combination with the high steric demand of the central $PtCl_2$ core. This hindered nucleation becomes more pronounced upon dilution and upon the addition of a supramolecular crowder. In order to keep the manuscript concise, we have chosen to include this thorough description in the supporting information as a supplementary note, whereas a brief discussion is added to the revised manuscript.

We also wish to point out that the observation of a kinetically hindered nucleation event has been widely reported in the literature, and this regime has been also ascribed to the molecularly dissolved state, which can be affected by conformational freedom or folding processes among other factors (please, for selected examples, see: *J. Am. Chem. Soc.* **2022**, *144*, 16, 7080, *Chem. Eur. J.* **2019**, *25*, 7303, *J. Am. Chem. Soc.* **2015**, *137*, 9, 3300.). We have provided this analogy and the appropriate citations in our revised manuscript.

2) As this study would open a new door in supramolecular polymer chemistry, I recommend this manuscript for publication in Nature Communications. However, the authors should discuss the mechanism at the molecular level with additional experimental evidences, particularly for the kinetic regime. In addition, as illustrated by GroES-GroEL complex shown in Figure 1a, it would be nice if the authors demonstrate that morphology 1 and morphology 2 of compound 1 are distinct in terms of their functions and properties, not only in terms of their morphology.

We appreciate the feedback and the suggestions from the referee along with their recommendation for publication. We have attempted to understand the mechanism at the molecular level in more detail by means of additional experimental results (please see above). While we agree that it would be of great interest to show the impact of supramolecular crowding on different (functional) properties other than the morphology, we would like to point out that our manuscript is a proof-of-principle example of crowding in supramolecular polymers. Currently, this concept is being applied to other dyes and self-assembled systems in our group, which may allow us to produce assemblies with different photophysical and electronic properties.

Reviewer 3: The paper titled “Biomimetic crowding directs supramolecular polymerizations” has claimed that the one-dimensional supramolecular polymers could act as an effective crowder and influence the kinetics, thermodynamics and morphology of the supramolecular polymerization processes of the coexisted monomers. In particular, the new morphology or the flower like assemblies can only be seen with one-dimensional supramolecular polymers as crowdors, not with spherical or sheet like supramolecular polymers or covalent polymers. It is an interesting manuscript that can be recommended for publication, provided some changes are made. Based on this, there are a few questions below.

We thank the referee for his/her encouraging comments and important feedback, which we have addressed as follows:

1. As the authors have compared the crowding effect in the supramolecular system with that in the biological system, could the authors relate the morphology of the crowdors to the excluded volume as what has been discussed in the biological system? For structures with smaller excluded volume, could higher concentrations promote the formation of the flower like assemblies? What's the molecular weight of the polystyrene tested in the paper? Could the polystyrene with a higher molecular weight promote the formation of the flower like assemblies?

We express our gratitude for this helpful suggestion. As previously discussed, we believe that the excluded volume does not have a significant contribution to the effects observed in our system (please see also response to referee 1 and 2). In order to probe a potential induction of a crowded environment by increasing concentrations of covalent (polystyrene) or spherical crowder (compound 6), we elevated the concentrations considerably to 10 mg/mL and 500 μ M respectively (in the case of compound 6 this is the highest possible concentration prior to precipitation). In neither case could the formation of flower-like assemblies be observed, highlighting that the fibrillar morphology of the crowder is the crucial characteristic (please, see image below). The polystyrene used in the experiments has a M_w of 13000 with $M_w/M_n = 1.06$. The micrographs and the additional information have been added to the corresponding section in the supporting information.

Figure 5: SEM micrographs of Agg1 obtained in the presence of highly elevated concentrations of **6** as a non-effective supramolecular (left) and polystyrene as a non-effective macromolecular (right) crowder using a concentration of 40×10^{-6} M of **1**.

2. Both the AFM and SEM are based on dried samples. How would the drying process influence the morphology? In the recently published paper “Hierarchically self-assembled homochiral helical microtoroids”, micrometer scale hierarchical structures are formed by drop casting the pre-assembled intermediate colloids in solution. Could the formation of the flower like structures in this paper have been caused by the slow evaporation process? Could the samples be spin coated and be compared with the drop casted samples?

We appreciate the helpful suggestion from the referee. Indeed, drying effects can play a major role in how supramolecular morphologies are visualized by techniques relying on the dried state. In order to exclude that our analysis is perturbed by drying effects, we have conducted a spin-rate dependent SEM study on Agg1U and Agg1C respectively, revealing that although the amount of structures on the surface is reduced upon increasing spin rates, the morphology is preserved (please see below). Additionally, we wish to point out that our light scattering experiments in solution also match the observed morphological properties in the dried state. The new micrographs have been added to the revised version of the supporting information.

Figure 6: Spin-rate dependent SEM analysis of Agg1U using a concentration of 80×10^{-6} M, revealing the stability of Agg1U against different drying processes.

Figure 7: Spin-rate dependent SEM analysis of Agg1C using a concentration of 80×10^{-6} M, revealing the stability of Agg1C against different drying processes.

3. In the paper by Zimmerman and Minton (*Annu. Rev. Biophys. Biomol. Struct.*, 1993), both of whom were cited by the authors, it was shown the reaction rate could first increase and then decrease with increasing amount of crowding reagent. If the concentration of the crowding reagent is reduced, could the acceleration of the supramolecular polymerization also be observed in addition to the delayed supramolecular polymerization observed in the paper?

We appreciate the helpful suggestion. Indeed, we have investigated this possibility for our system as well. Even at concentrations where compound **2** could not be clearly detected using UV/Vis analysis ($c < 0.5 \mu\text{M}$), the concentration was still high enough to affect the assembly behavior of **1** (please see below). We have added the corresponding spectra to the revised supporting information. Potentially, this originates from the differences in the molecular mechanism between our artificial and the natural systems described in the literature (see also response to referee 1 and 2).

Figure 8: Time-dependent UV/Vis spectra of mixtures of **1** and **2** following the kinetic self-assembly protocol described in the extended data at 40×10^{-6} M of **1** in the presence of increasing amounts of **2** (Top left to right: 0 and 0.5×10^{-6} M) with the absorbance at the absorption maximum ($\lambda = 350$ nm) plotted against the time.

4. In the abstract, the authors mentioned that “Within the growing field of supramolecular polymer science, crowding effects have hitherto not been considered”. The crowding effect is rarely explored in the supramolecular polymer science, but there are still papers studying the crowding effect of covalent polymers on the self-assembly behavior of PIC (ref 3).

We appreciate the referee for highlighting this aspect. We have modified the sentence in our revised manuscript to differentiate our work from the existing literature more clearly, highlighting the use of macromolecular crowders.

5. In Fig. 3a, the difference between Agg 1U and Agg 1C is small in the inset. Could an error bar be provided for both curves?

We thank the referee for this suggestion. In our initial submission, we have investigated this behavior at multiple concentrations to ensure that the trends are not artificial but instead originate from a reproducible phenomenon (please, see also extended data Fig. 3). To enable a statistical analysis, we have reproduced this experiment at a fixed concentration three separate times for crowded and uncrowded conditions (please see below). Our reinvestigation supported the conclusions made in our original concentration-dependent studies. Namely, it can be seen that all independent experiments show a higher hierarchical stability for Agg1C compared to Agg1U. In fact, even the first standard deviation does not show any overlap between the distinct aggregates for $0.1 < \alpha_{agg} < 0.9$, which is in line with previous concentration-dependent experiments. In order to not overload Fig. 3a, we have chosen to show this data in the revised supporting information, but have referred to it in the corresponding section of our manuscript.

Figure 9: Reproducibility study of the observed apparent increase in hierarchical stability using a concentration of $c_1 = 20 \times 10^{-6} \text{ M}$ (Agg1U) and $c_1 = 20 \times 10^{-6} \text{ M}$; $c_2 = 5 \times 10^{-6} \text{ M}$ (Agg1C).

6. Fig. 5 what are the functional groups (R) in a and b?

We thank the referee for highlighting this inconsistency. The R groups in 5a and b are the same as in 5c. We have modified the image to make this more visible.

REVIEWERS' COMMENTS

Reviewer #2 (Remarks to the Author):

The authors have addressed to the comments raised by the reviewers with additional experiments which are feasible at present. The finding of crowding effect in supramolecular polymerization is intriguing and will appeal the readership of Nature Communications. I recommend the manuscript for publication.

Reviewer #3 (Remarks to the Author):

The authors addressed most of the point adequately and I like to recommend publication in the form as it is now.